# Successful Treatment of Myofascial Pain Syndrome (MPS) with Surgical Cauterization of Temporalis Muscle Trigger Points: A Case Report

**DOI:** 10.3390/dj11010003

**Published:** 2022-12-23

**Authors:** Craig Pearl, Brendan Moxley, Andrew Perry, Nagi Demian

**Affiliations:** 1Department of Oral and Maxillofacial Surgery, The University of Texas Health Science Center at Houston, Houston, TX 77054, USA; 2School of Dentistry, The University of Texas Health Science Center at Houston, Houston, TX 77054, USA

**Keywords:** myofascial pain syndrome, myofascial trigger points, electrosurgery, temporomandibular disorder, pain relief, orofacial pain, headache

## Abstract

For patients suffering from myofascial pain syndrome (MPS) affecting muscles of mastication, traditional trigger point therapy treatment regimens can prove inconvenient, due to the short duration of pain relief after each injection and expense of repeated visits which are often not covered by insurance. We present a case of a patient treated using an alternative technique that could develop into an additional modality for treating MPS patients who are refractory to conservative treatment. This technique involves identifying and marking the patient’s trigger points and surgically cauterizing each location using a Bovie electrosurgical unit. While traditional trigger point injection therapy for myofascial pain syndrome is a well-described technique with acceptable pain relief expected for a period of 8–12 weeks, this technique provided up to 24 months of adequate pain relief in a patient. While further studies are indicated before widespread adoption can be recommended, this patient’s response suggests that this technique may be useful in offering longer-term pain relief compared with trigger point injection therapy.

## 1. Introduction

Myofascial pain syndrome (MPS) is a condition characterized by muscular trigger points with referred pain, muscle tenderness and spasms, and frequent headaches when affecting a muscle of mastication. These muscle spasms are typically caused by hyperexcitation of the peripheral sensory neurons which induces the muscle’s motor neurons. MPS pain in muscles of mastication can be debilitating, limiting normal daily activities, affecting the quality of sleep, and causing a myriad of other symptoms including decreased mouth opening, tension headaches, tinnitus, temporomandibular joint pain and cervical pain [1,2].

A myofascial taut band is best characterized as a contracted muscle fiber band with increased muscle tone that is painful to the touch. Trigger points are discrete, hyperirritable nodes within a taut band of skeletal muscle that can cause referred pain, tenderness, and autonomic phenomena beyond the site of palpation [3,4].

The specific technique used for treating MPS on the patient who is the subject of this report was modified from an existing technique that was proposed for the treatment of migraines without aura, in which extracranial trigger points were cauterized in the frontal, temporal, nasal and occipital sites [5]. Vincent et al. reported a success rate of 38% for this technique in treating migraines without aura which, though lower than ideal, they concluded remains a viable treatment due to the limited risk of complications [6]. The idea of utilizing thermal coagulation to treat chronic neuromuscular pain-related conditions is not new. Radiofrequency ablation and rhizotomy procedures have been used for treating cluster headaches and generalized myalgia. Both radiofrequency ablation (RFA) and pulsed radiofrequency treatment (PRF) of myofascial trigger points and scar neuromas in various body sites have shown high rates of success (89%) [7,8]. In this case, the thermal cauterization was performed using a Bovie electrosurgical unit (ESU) due to it being readily available in most oral and maxillofacial surgery clinics. 

When managing a patient with temporomandibular dysfunction, the initial step is typically to determine the origin of the pain which may be myogenic, arthrogenic or a combination of both. If the primary source of pain is determined to be myogenic, the next step would be to determine if the pain is localized (myalgia) or if it refers beyond its area of origin, as occurs in myofascial pain syndrome.

The management of chronic MPS has traditionally been very challenging, and with no universally agreed upon protocol, patients generally experience chronic pain, as the more traditional treatments only provide short term improvements with varied responses [9]. The initial conservative approach to treatment includes a combination of: home therapy with patient education, diet modification, occlusal splint therapy, physiotherapy and pharmacotherapy.

For patients who show no improvement in pain frequency or intensity after treatment with these conservative protocols, management would often then involve procedures such as diagnostic blocks with local anesthetic followed by trigger point therapy with botulinum injections. This treatment works especially well for patients who are prone to bruxism [10,11]. According to a double-blind, controlled placebo, randomized clinical trials with a six-month follow-up period, botulinum toxin type A injections resulted in an improvement of painful symptoms in up to 90% of patients, although the period of pain relief is limited to 8–12 weeks [12]. For many patients, botulinum toxin type A injections are all that they need to manage their chronic MPS. However, some disadvantages of this technique include the short duration of pain relief with each treatment, as well as the pain improving but not subsiding completely for many patients. Another disadvantage is the out-of-pocket cost to the patient, as this is often not covered by medical insurance. The patient may also require a higher dose than is traditionally used for cosmetic purposes, further increasing the cost [13].

This case report describes a patient who was initially successfully treated with diagnostic blocks with 2% lidocaine plain followed by trigger point injections with botulinum toxin A which provided him with pain relief for an average of 8 weeks between visits; however, he requested a more definitive, long-term pain relief solution to reduce the burden of monthly appointments and the number of workdays missed.

## 2. Presurgical Evaluation

This patient is a 49-year-old male who presented at the University of Texas Health Science Center at Houston with the complaint of a constant, deep, throbbing pain that started on the left temporal region and later progressed to a more diffuse pain on the left temple area. The patient reported that the pain began suddenly with no history of trauma or any other precipitating event that could account for the pain; the patient indicated that the pain was constant and had been present for five years. The pain was worse when supine at night and he rated it at 10/10 on the visual analog pain score. The patient denied that he clenched his teeth which seemed to be confirmed with his only mild occlusal wear. He mentioned that the pain affected his ability to sleep and participate in daily activities.

The patient was prescribed gabapentin by another practitioner who diagnosed the patient with neuralgia which was ineffective. The patient indicated that the amount of workdays missed was of great concern, either due to the severity of the pain itself or to appointments for treating the pain. He indicated in the last few years he has consistently missed a minimum of six weeks each year, putting his job and livelihood in jeopardy. In addition to the gabapentin, other treatment he received includes occlusal splint therapy, which he reported provided no improvement in the pain intensity nor frequency despite being compliant. He was also prescribed amitriptyline with very minimal benefit. The patient was diagnosed many years prior to the onset of the pain with a pituitary adenoma that did not require any intervention, only yearly follow-up at his neurologist. He has hypertension managed with hydrochlorothiazide, gastroesophageal reflux disease managed with omeprazole and prediabetes managed by diet alone.

The patient denies smoking, consuming alcohol or using drugs. His clinical exam was significant for marked tenderness over the left temporalis muscle with an extensive referral pattern beyond the area of palpation, primarily to the ocular, cervical and mandibular regions. He was noted to have a class 1 occlusion with unrestricted mouth opening of 55 mm inter-incisal distance as well as 10 mm right and left lateral excursion and 10 mm protrusion. On MRI, the patient had normal disc positioning both in the open and closed views as well as normal joint spaces bilaterally. The only significant finding was early degenerative changes of the left condyle with small subchondral cysts. Based on our clinical and radiographic examination, he was diagnosed with MPS as well as early degenerative changes in the left temporomandibular joint.

His first line of therapy was home education to limit his mouth opening to 30 mm, thereby limiting translation of the joints, to continue nightly wearing of the occlusal splint and to start low-dose cyclobenzaprine therapy (Flexeril 5 mg po nightly) as he was no longer taking his amitriptyline. With persistent pain on follow-up, diagnostic blocks were used to determine if the trigger points within the muscle were the cause of the patient’s pain. After palpating the left temporalis muscle, several trigger points were detected. The patient confirmed severe pain within the temporalis muscle as well as pain referring beyond the muscle, and he confirmed that the pain elicited by palpation was consistent with the pain he had been experiencing for the last 5 years. After injecting these trigger points with plain 2% lidocaine, the patient reported that he was pain free even beyond the areas where he was injected.

After obtaining informed consent, 30 units of botulinum toxin A were injected into the same six points that he received the lidocaine plain earlier. The patient returned after 3 weeks with only modest improvement of his pain. The patient indicated that he was feeling very despondent, as well as concerned about losing his job; therefore, we discussed the management of his condition with electrocautery.

## 3. Surgical Procedure

Materials: Bovie electrosurgery unit (Figure 1a), grounding pad (Figure 1b), a Colorado^®^ microdissection needle (Figure 1c), sterile draping, 2% lidocaine with epinephrine 1:100,000, sterile gauze, normal saline and alcohol or iodine wipes.

The general order for the procedure is (1) preparation, (2) local anesthesia, (3) normal saline, and (4) electrosurgery.

Preparation: The patient is positioned reclined in the supine position in an exam chair. If the area of interest is hair bearing, the area may be shaved, although this is not a prerequisite. The skin is prepared with any standard agent: alcohol, betadine, or chlorhexidine. Ensuring sterility, the areas of trigger points and taut bands are marked, as well as any anatomic landmarks delineated. In this case, the trigger points are marked with “X”s, the taut bands without pain referral with “O”s and the path of the temporal branch of the facial nerve and the auriculotemporal nerve with dashes (Figure 2a). A grounding pad is attached to the patient.

Local anesthesia: The local anesthetic is drawn into a 5 cc syringe using a sterile technique. This technique uses Lidocaine 2% with epinephrine 1:100,000. A 23-gauge needle is used to inject the local anesthetic into the skin and muscle. The needle is introduced perpendicular to the skin and is advanced until bony contact is made. The needle is then retracted slightly (~1 mm), and 0.5 cc of the local anesthetic is infiltrated (Figure 2b). Two to three minutes after the infiltration of the local anesthetic, the accuracy and precision of your initial trigger point marking is confirmed with complete relief of pain; if the patient still reports pain in that area or referred from that site then additional points may be marked and anesthetized.

Normal saline: Normal saline is drawn into a 5 cc syringe using a sterile technique. An 18-gauge needle is used to inject the sterile saline into the anesthetized, marked trigger point. Like the injection of local anesthesia, the needle is introduced perpendicular to the skin, and is advanced until bony contact is made. The needle is retracted slightly (~1 mm), and 1 cc of the saline is deposited (Figure 3a). Muscle is a good conductor of electricity with low resistance, so infiltrating NS into the tissue increases the resistance but decreases the conduction of electricity, thereby decreasing the heat generated and distributing it over a larger surface area [14].

Electrosurgery: An electrosurgical unit (in this case a Bovie) is prepared with a Colorado needle tip (Stryker Colorado Needle). The Bovie is set to a power of between 15–20 watts on the blend setting (50%/50% split between cut and coagulate). The tip of the Bovie needle is introduced perpendicular to the skin through the same hole that was created when injecting the 1 cc of saline, and is advanced beyond the skin into the area where the injected saline is seen to be tenting up the skin. The Colorado needle tip should not hit against the bone to avoid damaging the tip (~1 mm); the electrosurgical unit is then activated on coagulate for 5 s (Figure 3b).

The patient underwent surgical cauterization of his trigger points according to the aforementioned technique. To date, he has had five sessions of surgical cauterization with complete pain cessation experienced for a minimum of 2 months and a maximum of 2 years between visits (Figure 4). 

In the first electrosurgery appointment, we were able to locate 6 trigger points when palpating the left temporalis muscle, all with a referral pattern of pain. The patient responded in an extreme manner when these trigger points were palpated, but there were no trigger points in any of the other muscles of mastication bilaterally. These six trigger points were then treated per the technique described previously (Figure 2c,d). The patient tolerated the procedure well, and by his post-operative appointment 3 weeks later, he reported a 0/10 visual analog pain score (VAS) for pain and no headaches or missed workdays. He only experienced 24 h of tenderness over the points that were treated.

The patient returned for a routine follow up appointment two months after the initial treatment with electrocautery. He reported relief from pain in all six sites; however, he did report becoming aware of additional tender points in his left temporalis muscle rated at 5/10. Of significance, we did identify the precise points the patient was mentioning 2 months prior, although they did not present as trigger points at the time, but rather as taut bands, as they were mildly painful on palpation with no referred pattern of pain. The five additional, though less severe, points were injected with 2% lidocaine with epinephrine 1:100,000 and were treated in a similar fashion by Bovie electrosurgery on a Colorado tip on blend, to a setting of 20 watts for 5 s. As this is an evolving technique, we determined that because the Colorado tip affects less than 0.18 mm of tissue, we increased the power to 20 watts for 5 s. This treatment provided 4 months of complete pain relief, a duration that exceeded any previous treatment for this patient, irrespective of the treatment modality.

After 4 months, the patient returned as he felt that his original pain was slowly returning. For 4 months, his pain was 0–1/10, but now was a 7/10 based on the visual analog pain score. Again, all the points identified were in different locations to the trigger points and taut bands without pain referral that we initially treated. Upon palpation, only 3 trigger points were identified with their typical referred pain pattern. Once again, we performed diagnostic blocks with 2% Lidocaine with epinephrine 1:100,000, which the patient confirmed eliminated the pain completely. The trigger points were then injected with 1 cc of normal saline, followed by electrosurgery with a Bovie needle.

The patient did not attend his 2–3 week follow-up due to work-related issues; however, he did indicate that he was completely pain free with no need for medication, and informed us that he wished to stop taking his gabapentin. We contacted his neurologist, who indicated he would see the patient and discontinue the medication if the patient’s pain had improved. This treatment provided 24 months of complete pain relief between 2020 and 2022.

The patient returned in February 2022, indicating that the pain had only started returning in December 2021, which he rated as a 5/10 on the visual analog pain score. A total of five trigger points over the left temporalis muscle were identified, which once again appeared to be in a different position on the muscle from the sites that were previously treated. The patient was managed in a similar manner as he was previously, with diagnostic blocks confirming the immediate pain relief, followed by electrosurgery of the five diagnosed trigger points.

At the post-operative appointments 2–3 weeks after the procedure, the patient reported only minor discomfort for 24–48 h. The patient reported that he is sleeping better without the need for additional medication. The patient also inquired if he could stop using his occlusal splint at night; however, we strongly urged him to continue wearing it. The patient also reported that he felt numb in two areas over the left temporalis muscle. However, clinically, the patient exhibited normal response to touch, pain, temperature and two-point discrimination when examined. 

The patient returned after a 3 month period, indicating that the pain was no more than 4/10 on the VAS and was confined to two or three spots. On examination, three points were identified with a referred pain pattern, but were in no way as severe as all the previous points identified. Importantly, these three points were once again in different positions to the previously treated trigger points, although they were in close proximity to previously identified tender points on the taut bands without trigger points. The patient also stated that the sensation over the left temporalis muscle was normal, which was then confirmed clinically. A numerical representation of the patient’s treatment follows (see Figure 4 below). 

**Figure 4 dentistry-11-00003-f004:**
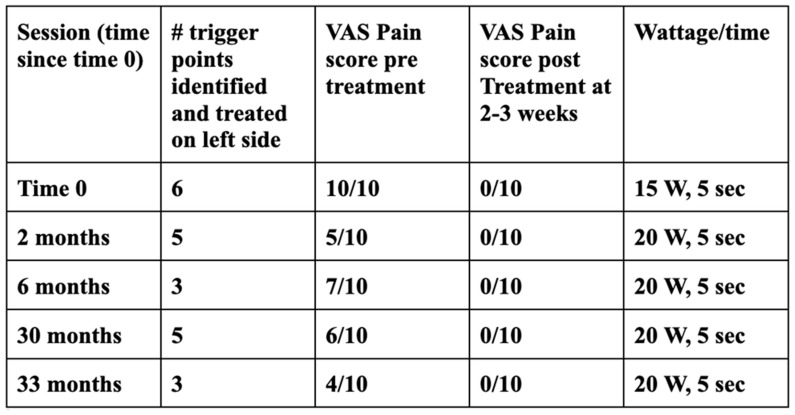
The patient received a total of five treatments. The important points from each visit are outlined above.

## 4. Post-Operative Outcome

At a post-operative appointment 2 weeks after the fifth and final electrosurgical cauterization procedure, the patient reported to be doing well, and indicated that he was completely pain-free and had discontinued his gabapentin since the last procedure following his appointment with the neurologist. He also reported no altered sensation over the overlying skin. The patient indicated that his quality of life had improved since we started treating his trigger points with electrocautery. He has not returned for 8 months since the last procedure (see Figure 4 for timeline). 

## 5. Discussion

A Pubmed search conducted in November of 2022, using MeSH terms including “myofascial pain syndrome”, “myofascial trigger points” and “electrosurgery” did not reveal a precedence of treating MPS with electrosurgery of the temporalis muscle, nor any other muscle of mastication. While further studies with larger numbers of patients are necessary to determine the efficacy of this treatment, subjective reporting from the patient is encouraging. Chronic MPS patients typically are only afforded 2–3 months of pain relief from botulinum toxin A trigger point injection therapy, with a variable efficacy between each trigger point therapy. With this novel technique, however, the patient experienced adequate pain reduction for 24 months after the third trigger point electrosurgery session. The literature review described trigger point injection techniques with durations of pain relief much shorter than the 2 years afforded between treatments 3 and 4, and while the electrosurgery results were variable in duration, they were consistent in effectiveness of pain relief.

One of the most important steps in managing trigger points is accurately identifying and separating the taut bands containing trigger points from those without referred pain. Once identified, trigger points should be injected with local anesthetic to determine the exact nature of the referred pain. If the pain stems from trigger points, it should improve after management, irrespective of treatment technique. In our experience, treating the trigger points on the same day as the diagnostic blocks has improved the accuracy and effectiveness of trigger point therapy with both botulinum toxin A injections and electrocautery. One interesting finding was that after the initial treatment of the identified trigger points, the taut bands without pain referral initially identified now presented as trigger points, 2 months later. The significance of this has yet to be determined, especially when determining if this would become a frequent occurrence when managing trigger points with the electrocautery technique.

As previously stated, muscle is a good conductor of electricity with low resistance, which results in a larger area of muscle being affected by electrosurgery [14]. Trigger points are estimated to have a surface area of about 1 cm^2^. Thus, based on our assumption of the appropriate volume, we estimated that 1 cc of sterile saline should be initially injected into each of the trigger points to decrease the conductivity of electricity with an increased resistance, resulting in a lower temperature over a smaller surface area, thereby improving the accuracy and effectiveness of the electrosurgery [15]. This technique, known as saline-coupled bipolar electrocautery, is a well-established technique used in various medical specialties for low-temperature coagulation, which in this case prevents necrosis of the skin or damage to the hair follicles [16,17]. 

Another important step to improve accuracy and reduce potential complications is the choice of Bovie tip utilized, which in this case was the Colorado^®^ microdissection needle which has an ultra-sharp tungsten tip that delivers the wave-form from the electrosurgery generator to a very small spot (0.18 mm). This allows the use of extremely low wattages, resulting in decreased tissue necrosis, precision cutting and cautery and reduced postoperative pain. Because this procedure is performed deep to the epidermis, dermis and subcutaneous connective tissue, but is within the muscle layer, the risk of burning skin and hair follicles is eliminated. The patient did not suffer any hair loss after the electrocautery procedures, even after the patient had five treatments over the three-year period. By ensuring that the Colorado needle tip passes down into the deeper layer prior to using the tip, we can not only prevent hair loss, but also minimize scarring or altered pigmentation. This is why this technique describes using an 18-gauge needle to inject the NS, which provides a smaller puncture wound suitable to insert the Colorado needle tip without affecting the dermis and epidermis, and also prevents damaging the needle tip.

Trigger points can form when a muscle band contracts for an indefinite period of time, causing the muscle to switch to anaerobic respiration, resulting in lactic acid buildup and pain when released upon palpation or exertion by the patient. In addition, these muscular changes are coupled with biochemical changes, including increased pro-inflammatory molecules such as substance P, IL-1β, and tumor necrosis factor (TNF)-α which activate the muscle’s nociceptors [18]. The referred pain characteristic of MPS can be explained by peripheral and central sensitization, in which constant nociceptor activation overwhelms neuromodulator release, resulting in an amplification of synaptic communications between the first- and second-order neurons [19]. Then, the recruitment of non-nociceptive fibers in non-affected areas that synapse on the same second-order neurons causes referred pain [19]. A proposed route of effect of the intervention described in this case report is that by replacing the muscle associated with the trigger point with fibrous scar tissue, the skeletal muscle in the initial trigger point area can no longer contract, reducing the biochemical concentrations of substance P, IL-1β, and tumor necrosis factor (TNF)-α and eliminating the referred pain. The effect of this technique is similar to that of traditional BTX-A injections which temporarily paralyze the trigger point; although, electrosurgical cauterization results in much longer-lasting paralysis of the trigger point.

Besides the 4-week period of paresthesia over two sites, the patient did not express any associated morbidity with the procedure; his mastication was unaffected and he showed minimal to no scarring. The procedure was easily performed under local anesthetic, although the method of anesthesia would depend more on the type of patient rather than the procedure. Regardless, it is important to ensure that the trigger points are located and marked prior to the patient receiving any medication that may lead to drowsiness. 

## 6. Conclusions

For this patient, this non-traditional treatment method was more successful than any previous conservative therapy or trigger point injections. While there was still a recurrence of the pain for the patient, albeit in different places on the muscle, the periods between treatments were longer than any of the traditional treatments had offered. Furthermore, despite the patient returning for continued treatment, the pain was always associated with less intensity and the patient always expressed a general improvement in quality of life as his treatment progressed. This is a case report with ample limitations, but the results noted by this patient may indicate that this is an area that is worthwhile for future study. 

## Figures and Tables

**Figure 1 dentistry-11-00003-f001:**
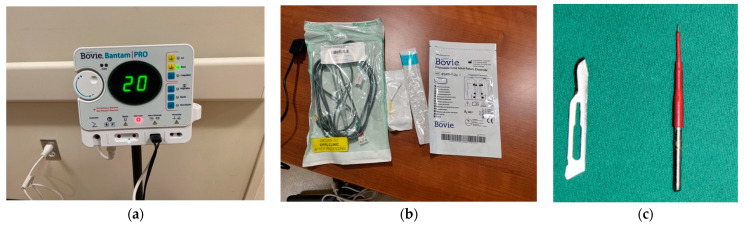
(**a**): A Bovie ESU set to 20 watts. (**b**): The wiring and electrosurgical tip for the Bovie. (**c**): A Colorado tip next to a #15 scalpel blade for size reference.

**Figure 2 dentistry-11-00003-f002:**
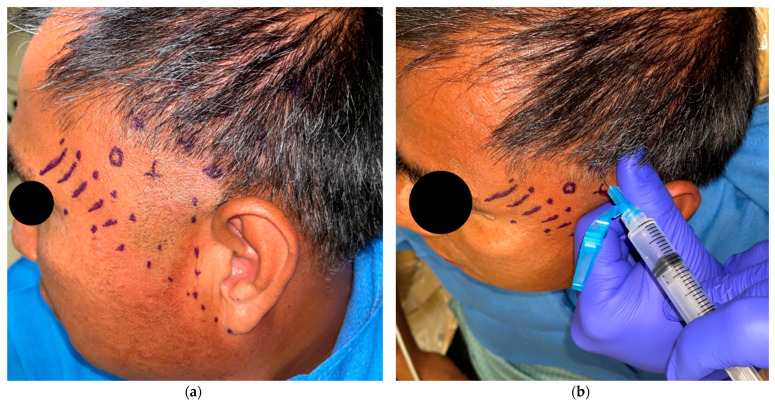
(**a**): Trigger points (X), Taut bands without referral (O), dashed lines = predicted path of facial nerve. (**b**): Injection of local anesthetic into each trigger point.

**Figure 3 dentistry-11-00003-f003:**
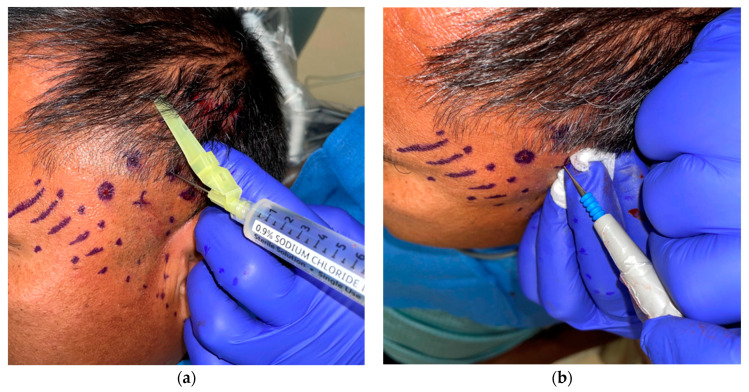
(**a**): 0.9% normal saline is injected into each trigger point using an 18 G needle. (**b**): Electrosurgery is performed on each trigger point for 5 s to a setting of 20 watts on coagulate.

## Data Availability

Not applicable.

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
