# Peer review of "Successful Treatment of Myofascial Pain Syndrome (MPS) with Surgical Cauterization of Temporalis Muscle Trigger Points: A Case Report"

_dentistry, 2022, doi:10.3390/dj11010003_

Round 1
Reviewer 1 Report
The manuscript is well written and the topic is interesting; but there are some points that must be improved:
1) the case must to be presented in a different way: presurgical evaluation; surgical procedure; post-operative outcome;
2) the Authors must to provide patient's images and photos (preoperative and postoperative clinical exam, with eventually mouth opening, surgical procedure)
3) It is preferable to replace older literature with newer articles
Author Response
Cover letter for Reviewer 1
- The presentation of the case has been renamed to reflect the presurgical evaluation, surgical procedure, and post-operative outcome sections. A separate reviewer also made a comment about re-organizing the presentation of the paper, although the other reviewer did not specify the 3 sections.
- As for the patient’s images and photos, the only photos we have of the patient are those that are included in the surgical procedure section. This patient was treated initially without the intention of writing a case report. It was only after the patient responded so well to this non-traditional treatment method that it was decided to write a case report. Thus, patient images are lacking. Furthermore, as for mouth opening, it is mentioned in our paper that “He was noted to have a class 1 occlusion with unrestricted mouth opening of 55 mm inter-incisal distance as well as 10 mm right and left lateral excursion and 10 mm protrusion.” Thus, for this patient, the mouth opening was not a concern, which is even more reason why we wouldn’t have pictures of this type.
- Admittedly, some of our sources are a bit older. However, this is in part due to the subject material. According to a 2020 paper published in Springer entitled “A comprehensive review of the treatment and management of myofascial pain syndrome, ”MPS is poorly understood and remains a challenging condition to treat. Non-pharmacologic treatment modalities such as acupuncture, massage, transcutaneous electrical stimulation, and interferential current therapy may offer relief to some patients with MPS. Additional studies are warranted to get a better understanding of managing myofascial pain.” In short, many of the mechanisms associated with referred pain are not clear, and this specific modality described in this case report is not described in the literature for use in treating MPS. Thus, there are not very many new sources to cite for this specific case report. However, the 3 sources that were added in making other revisions to the paper have all been from within the last 2 years.
Reviewer 2 Report
No suggestions
Author Response
Cover letter for reviewer 2:
- The only thing that this reviewer mentioned was that the English was very difficult to understand or was incomprehensible. The authors of this paper are all native English speakers and the other two reviewers did not make more than minor spell-check suggestions.
Reviewer 3 Report
Dear editor
Thank you for inviting me to review the manuscript. This is a well-written article and I think can be accepted after some revisions.
1- The format of the article is not appropriate for a case report. Please revise the whole manuscript.
2- The conclusion should be only based on the results of your own study.
3- In the discussion please add some reasons for the possible routes of the effect of your intervention on pain relief.
Author Response
Cover letter for reviewer 3:
- One of the other reviewers also made the suggestion that we re-format the manuscript to better fit an appropriate case-report presentation. The other reviewer specifically suggested that the paper be broken up into 3 sections: Presurgical evaluation, Surgical Procedure, and Post-operative outcome. The paper has been structured accordingly.
- The conclusion section has been revised to only indicate the patient’s results. I kept the final statement about how this may be a worthwhile area of future study, as that is what is needed before this technique can be recommended. As this is a case report, our goal in publishing it is not to change anyone’s way of practice, nor to suggest that it will be effective for other. It may or may not be; we cannot say. Rather, we hope to stimulate thought and discussion in this area.
- The second-to-last paragraph of the discussion has been revised to include a possible route of effect of this intervention on pain relief.
Round 2
Reviewer 1 Report
The paper, after these changes, can be accepted